# Lactose Permease Scrambles Phospholipids

**DOI:** 10.3390/biology12111367

**Published:** 2023-10-25

**Authors:** Lei Wang, Peter Bütikofer

**Affiliations:** Institute of Biochemistry and Molecular Medicine, University of Bern, 3012 Bern, Switzerland

**Keywords:** LacY, scramblase, phospholipids, bilayer, symporter

## Abstract

**Simple Summary:**

Scramblases are proteins that translocate phospholipids from one leaflet of a membrane bilayer to the other. Our work identifies that a well-characterized protein from *Escherichia coli*, lactose permease, also displays lipid scrambling activity. The scrambling function is independent of LacY’s proton-coupled lactose transport activity and involves two amino acid residues located near the membrane surface.

**Abstract:**

Lactose permease (LacY) from *Escherichia coli* belongs to the major facilitator superfamily. It facilitates the co-transport of β-galactosides, including lactose, into cells by using a proton gradient towards the cell. We now show that LacY is capable of scrambling glycerophospholipids across a membrane. We found that purified LacY reconstituted into liposomes at various protein to lipid ratios catalyzed the rapid translocation of fluorescently labeled and radiolabeled glycerophospholipids across the proteoliposome membrane bilayer. The use of LacY mutant proteins unable to transport lactose revealed that glycerophospholipid scrambling was independent of H^+^/lactose transport activity. Unexpectedly, in a LacY double mutant locked into an occluded conformation glycerophospholipid, scrambling activity was largely inhibited. The corresponding single mutants revealed the importance of amino acids G46 and G262 for glycerophospholipid scrambling of LacY.

## 1. Introduction

Glycerophospholipids constitute indispensable constituents of biological membranes. In both eukaryotic and procaryotic cells, phosphatidylethanolamine (PE) and phosphatidylcholine (PC) are among the most abundant classes. They can be synthesized de novo via the two branches of the Kennedy pathway located in the cytosol and on the cytosolic surface of the endoplasmic reticulum (ER) in eukaryotic cells and in the cytoplasm and on the cytoplasmic side of the cell membrane in prokaryotic cells [1,2,3,4]. Insertion of newly synthesized PE and PC into the outer leaflet of the ER and inner leaflet of the cell membrane, respectively, results in their asymmetric arrangement between the two halves of the membranes. Thus, to rectify or maintain the stability of the bilayer structure, rapid transfer of the newly synthesized lipids to the opposing membrane leaflets has to occur. However, due to the unfavorable energy requirement associated with the spontaneous translocation of glycerophospholipids across a bilayer, this process occurs at a relatively slow rate, with half-lives reaching up to 100 h in a fluid PC bilayer. Therefore, specific proteins with catalytic capabilities are required to facilitate this translocation process. Among these transport proteins, some translocate lipids unidirectionally across the bilayer using ATP as their energy source, while others enable bidirectional and energy-independent transport of lipids [5,6,7,8,9].

Glycerophospholipid scrambling across lipid bilayers has been studied for more than thirty years; however, proteins with scrambling activities have been difficult to identify. [10,11,12]. Only recently have several glycerophospholipid scramblases been characterized on a protein level. These include TEME16F, a central player in the initiation of blood coagulation and the fusion of trophoblasts [13,14]; CLPTM1L (cleft lip and palate transmembrane protein 1-like), a lipid scramblase involved in the cytosol-to-lumen translocation of glucosaminyl-phosphatidylinositol for glycosylphosphatidylinositol biosynthesis [15]; rhodopsin and its apoprotein opsin, a photoreceptor of vision [16,17,18]; Xkr8 (XK-related protein 8), which exposes phosphatidylserine on the cell surface of mammalian cells during apoptosis [19].

Lactose permease (LacY) of *Escherichia coli* is the prototype of the major facilitator superfamily (MFS). It is encoded by the lacY gene which represents the second structural gene in the lac operon [20,21]. LacY belongs to the oligosaccharide/H^+^ symporter subgroup of the MFS and is the sole transporter of β-galactosides in *E. coli*. LacY recognizes disaccharides composed of a D-galactopyranosyl ring and D-galactose and exhibits no affinity for glucopyranosides or glucose. The primary substrate of LacY is lactose, which is co-transported with a proton [20]. LacY was the first membrane transport protein to be cloned, sequenced, purified and structurally characterized [22,23,24]; it consists of 417 amino acid residues, of which 65–70% are hydrophobic, and is structured as a pair of hexahelical bundles that are linked by a relatively extended cytoplasmic loop. LacY is a monomer in the cell membrane and reconstitutes as a monomer when incorporated into liposomes [23]. 

A prevalent model for explaining the membrane transport of lactose is the alternating access mechanism [25,26], wherein H^+^ symporters cyclically present H^+^- and cargo-binding sites to either side of the membrane by undergoing conformational alterations. In the case of LacY, both sites for sugar binding and H^+^ binding are positioned near the center of the molecule, precisely at the apex of the cavity. This spatial arrangement serves as the structural foundation that supports the functioning of the alternating access mechanism [25,26].

In this study, we show that LacY reconstituted into liposomes is capable of scrambling glycerophospholipids across the membrane and that lipid scrambling is independent of H^+^/lactose transport activity. Amino acid residues G46 and G262 of LacY were identified as key residues for scramblase activity.

## 2. Materials and Methods

### 2.1. Chemicals and Reagents

Unless otherwise stated, all reagents were of analytical grade and purchased from Sigma–Aldrich or Merck. Phosphatidylcholine and *E. coli* Extract Polar, 1,2-dioleoyl-sn-glycero-3-phospho-(1’-rac-glycerol) (DOPG) and 7-nitro-2,1,3-benzoxadiazol (NBD)-labeled glycerophospholipids were purchased from Avanti Polar Lipids Inc. (Alabaster, AL). L-α-[*myo*-Inositol-2-3H(N)] phosphatidylinositol ([^3^H]PI; 11.8 Ci/mmol) was purchased from American Radiolabeled Chemicals, Inc. (St. Louis, MO). PI-specific phospholipase C (PI-PLC) from *Bacillus cereus* was purchased from Thermo Fisher Scientific. SM-2 Bio-Beads adsorbents were obtained from Bio-Rad. Triton X-100 was purchased from Roche Applied Science. n-Dodecyl-β-D-maltopyranoside (DDM) was purchased from Anatrace (Berkshire, UK). NBD-glucose was purchased from Avanti Polar Lipids.

### 2.2. Protein Expression and Purification of LacY and Its Mutants

LacY and its mutants were transformed into BL21 *E. coli* cells and overexpressed as described elsewhere [27]. Briefly, the cells were grown in LB medium supplemented with 0.1 mg/mL ampicillin. Protein expression was induced with isopropyl-β-D-thiogalactopyranoside and incubated at 16 °C for 3 h, then grown at 37 °C overnight. Cells were then harvested and resuspended in lysis buffer (20 mM Tris·HCl pH 8.0, 500 mM NaCl) supplemented with 5 mM EDTA, and membrane aliquots were frozen in liquid nitrogen and stored at −80 °C until further use. The lysate was incubated for 2 h at 4 °C in 20 mM Tris·HCl pH 8.0 containing 300 mM NaCl, 10% (*v*/*v*) glycerol and 4% (*w*/*v*) n-nonyl-β-D-glucopyranoside (buffer A) for purification. After ultracentrifugation (100,000× *g* for 1 h at 4 °C), the supernatant was diluted in the same buffer and incubated with 0.5 mL (bed volume) pre-equilibrated Ni-NTA Superflow resin (Qiagen) for 2 h at 4 °C. Then the resin was transferred into a column and washed three times with 5 mL buffer A using gravity flow. Finally, the resin was eluted with lysis buffer as previously described [27].

### 2.3. Reconstitution of LacY into Liposomes

LacY-containing proteoliposomes were prepared as described previously [28]. Briefly, preformed liposomes composed of ePC and *E. coli* lipid extracts were stabilized with detergent, and proteins of interest were added. Subsequently, the detergent was slowly removed using incubation with Bio-Beads. Protein-free liposomes were prepared in parallel.

### 2.4. PI-PLC Assay

The assay was performed as previously described [28]. Briefly, PI-PLC (3 µL) was added to (proteo)liposomes followed by rapid mixing. After incubation at room temperature for 0–10 min as indicated in the experiments, the enzyme reaction was terminated by transferring the samples to a glass tube containing 0.5 mL chloroform and 1.0 mL methanol. Lipids were extracted by adding distilled water and frequent vortexing for 10 min at room temperature. After the addition of 0.5 mL chloroform and 0.5 mL methanol, the resulting two-phase extract was vortexed and then centrifuged for 5 min at 2500 rpm using a Universal 2S centrifuge (Hettich, Tuttlingen, Germany). The lower organic phase, containing nonhydrolyzed [^3^H]PI, and the upper aqueous phase, containing [^3^H]inositol-cyclic-phosphate, were transferred to separate scintillation vials and counted in a liquid scintillation counter.

### 2.5. Scrambling of NBD-Lipids

Scrambling of NBD-PC was measured as described before [17,28,29]. Briefly, liposomes or proteoliposomes composed of NBD-lipids and other lipids were, as indicated, diluted with HEPES buffer, and NBD fluorescence was monitored in a fluorimeter until a stable signal was obtained. Bleaching of the fluorescence in the outer leaflet of liposomes was achieved with the addition of dithionite.

## 3. Results and Discussion

### 3.1. LacY Is a Phospholipid Scramblase

To study phospholipid scramblase activity, LacY was purified and reconstituted into liposomes consisting of ePC and *E. coli* polar lipid extract and containing trace amounts of fluorescent NBD-labelled glycerophospholipids as probes. This assay has been successfully used before to identify and characterize novel scramblases [18,30,31,32,33,34]. It measures the time course of fluorescence loss upon the addition of membrane-impermeant dithionite, reaching a 50% reduction in protein-free liposomes and higher values in proteoliposomes containing scramblase activity [28,29] (Figure 1a).

LacY protein from *E. coli* was overexpressed and purified with an apparent molecular mass of ~34 kDa (Figure 1b and Appendix A). NBD-PC, with the fluorescent group linked to the sn-2 position of the glycerol, was used as a probe to measure glycerophospholipid scrambling. We observed ~65% and ~85% fluorescence loss after the addition of dithionite in proteoliposomes reconstituted with LacY at protein to phospholipid ratios (PPRs) of 0.5 and 2.0 mg/mmol and ~50% reduction in protein-free liposomes (Figure 1c). The fluorescence decay in proteoliposomes reconstituted at a PPR of 2 mg/mmol reached a plateau approximately 6 min after the addition of dithionite and could be best described by a one-phase exponential function with ~t1/2 of 8.8 ± 0.1 s (n = 3). Similar results were obtained using N-NBD-PE, with the fluorescent group attached to the phospholipid head group. At PPRs of 0.5 and 2.0 mg/mmol, fluorescence reductions were ~67% and ~85%, respectively, with a fluorescence decay (t1/2) of 12.3 ± 0.2 s (n = 3) at a PPR of 2.0 mg/mmol (Figure 1d). In addition, reconstituted LacY also scrambled NBD-PI, with the fluorescent group linked to the sn-2 position of the glycerol. At PPRs of 0.5 and 2.0 mg/mmol, ~65% and ~85% of NBD-PI were reduced by dithionite, with a t1/2 of 12.8 ± 0.2 s (n = 3) at a PPR of 2.0 mg/mmol (Figure 1e). When the amount of LacY used for reconstitution varied between 0.25 mg/mmol and 4 mg/mmol, the extent of NBD-PC reduction increased from ~60% to ~90% (Figure 1f). Taken together, the data show that LacY reconstituted into liposomes scrambles NBD-labeled glycerophospholipids.

To rule out the possibility that the increased fluorescence loss in proteoliposomes may be caused by dithionite permeation in LacY-containing proteoliposomes rather than from LacY scrambling activity, NBD-labeled glucose leaking assays were performed. First, NBD-labeled glucose was encapsulated into (proteo)liposomes (Appendix A). The fluorescence of NBD-glucose was monitored after the addition of dithionite over time. The results show that NBD-glucose was reduced at similar rates in protein-free and LacY-containing (proteo)liposomes (Appendix A). The initial drop in fluorescence trace right after the addition of dithionite results from a small amount of NBD-glucose present on the outside of the (proteo)liposomes (Appendix A). The NBD-glucose leaking assay showed that dithionite does not penetrate LacY proteoliposomes during the time scale of the experiment.

In addition to using NBD-labeled glycerophospholipids as probes, we also measured LacY-mediated scrambling of the natural ^3^H-labeled glycerophospholipid, [^3^H]PI, using PI-PLC as a probe [28] (Figure 2a). This method has been used before to determine the bilayer organization of PI and derivatives and to characterize novel scramblases [15,28,35,36,37]. Our results show that PI-PLC treatment of [^3^H]PI-containing proteoliposomes reconstituted with LacY at PPRs of 2, 4 and 9 mg/mmol resulted in ~62%, ~75% and ~82% hydrolyses, respectively, compared to ~50% hydrolysis in protein-free liposomes (Figure 2b). Taken together, our results demonstrate that purified LacY reconstituted into liposomes mediates the scrambling of NBD-labeled and natural glycerophospholipids across a membrane.

### 3.2. Glycerophospholipid Scrambling by LacY Is Independent of H^+^/Lactose Transport

To study if the glycerophospholipid scrambling of LacY is linked to H^+^/lactose transport activity, we first determined NBD-PC scrambling by LacY in the absence and presence of lactose at different pH values. The results show that scrambling of NBD-PC was not affected by the presence of lactose and/or at increased proton concentrations (Figure 3a,b). Second, we expressed several LacY mutants containing point mutations known to block lactose transport [20]. The mutated amino acids are all located in the aqueous cavity of LacY, with Glu126 (in helix IV) and Arg144 (in helix V) being involved in substrate binding and Glu269 (in helix VIII) in coupling proton translocation and sugar binding (Figure 3c). The mutated forms of LacY (E126A, R144A and E269A) were purified (Figure 3d and Appendix A) and subsequently reconstituted individually into liposomes. Glycerophospholipid scrambling was measured using NBD-PC and [^3^H]PI as probes. The results showed no differences in the fluorescence reduction in NBD-PC (Figure 3e) or hydrolysis of [^3^H]PI by PI-PLC (Figure 3f) between wild-type and LacY mutants E126A, R144A and E269A.

Together, these results demonstrate that glycerophospholipid scrambling by LacY is independent of its H^+^/lactose transport activity.

### 3.3. Occluded Conformation of LacY Blocks Glycerophospholipid Scrambling

It is well established that certain mutants of LacY show altered protein conformations. To study if changes in the protein conformation may affect LacY glycerophospholipid scrambling activity, we first generated the LacY mutant D68N, in which the protein is locked in a cavity-inward-facing conformation [38] (Figure 4a). After expression and purification of the mutant protein (Figure 4b and Appendix A), it was reconstituted into liposomes and analyzed for glycerophospholipid scramblase activity using NBD-PC and NBD-PE as probes. The results shown in Figure 4c,d reveal that the D68N mutant had identical scramblase activity as wild-type LacY. Second, we studied the LacY double mutant G46W/G262W, which has been shown to lock into a partially open outward-facing conformation when crystallized in the presence of β-D-galactopyranosyl-1-thio-β-D-galactopyranoside (TDG), with the cytoplasmic side being tightly sealed [39] (Figure 5a). The opening on the periplasmic side is too narrow to allow the entrance or exit of sugar substrates; however, the double mutant is able to bind sugar substrates and initiate the transition but cannot complete it into an occluded state as the bulky Trp residues at positions 46 and 262 block complete closure [20,39]. Unexpectedly, we found that when the purified LacY mutant G46W/G262W (Figure 5b and Appendix A) was reconstituted into proteoliposomes in the presence of 1 mM TDG at a PPR of 2 mg/mmol, scrambling of NBD-PC was drastically reduced, i.e., from 85% for wild-type LacY to 57% for LacY mutant G46W/G262W (Figure 5c). A similar result was obtained with N-NBD-PE as a probe to measure glycerophospholipid scrambling (Figure 5d). These results indicate that the locked outward-facing conformation, or the mutated amino acids, inhibit the scrambling of NBD-glycerophospholipids.

### 3.4. Amino Acid Residues G46 and G262 Are Important for LacY Scramblase Activity

Since the LacY double mutant G46W/G262W showed no glycerophospholipid scramblase activity, we studied if the corresponding single mutants (G46W or G262W) also affected scrambling. After expression, purification (Figure 6a,c and Appendix A) and individual reconstitution of the mutants into liposomes, we observed that NBD-PC scrambling was partially inhibited by either mutant (Figure 6b,d), indicating that residues G46 and G262 of LacY are crucial for glycerophospholipid scrambling activity. Interestingly, the conserved residues G46 and G262, which are located in transmembrane helices 2 and 8, respectively, are near the protein-phospholipid interface on the extracellular surface (Figure 6a). It is tempting to speculate that they may be involved in modulating phospholipid binding or lipid scrambling.

## 4. Conclusions and Perspectives

Our work reveals that purified and reconstituted LacY from *E. coli* is able to scramble glycerophospholipids across the membrane in proteoliposomes and that this activity is independent of LacY’s H^+^/lactose transport activity. In addition, we identified two amino acid residues that are crucial for LacY scramblase activity. Other (point) mutations of LacY affecting H^+^/lactose transport, or protein structure, may help delineate the mechanism of lipid scrambling.

Our work does not address the functional relevance of LacY lipid scrambling. However, it should be noted that in most cases reported so far, lipid scrambling activity has been identified in proteins known for other functions, such as several members of the TMEM16 and XK families and multiple G-protein-coupled receptors [14,16,18,19,31,32,40], i.e., lipid scrambling can be considered a moonlighting function of (well-known) membrane proteins. We speculate that other members of the MFS, to which LacY belongs, may also display scrambling activity.

The molecular mechanisms underlying protein-mediated lipid transport are not well understood. A general feature of lipid scramblases seems to be the existence of a hydrophilic groove that faces the hydrophobic core of the lipid bilayer, as proposed in a model more than 15 years ago [5]. In addition, scramblases may promote lipid transport by locally thinning the bilayer [41,42]. Interestingly, very recently, several members of a protein family that shares these features, the membrane protein insertases, have been shown to display lipid scrambling activity [37,43,44].

## Figures and Tables

**Figure 1 biology-12-01367-f001:**
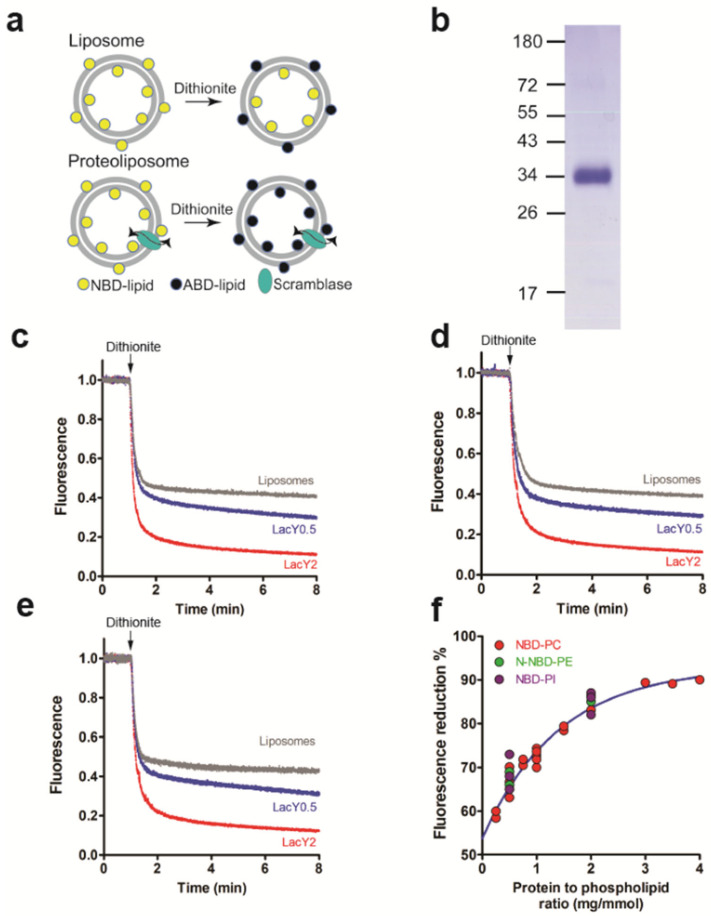
LacY scrambles phospholipids. (**a**) Schematic representation of scramblase assay. NBD-labeled glycerophospholipids (NBD-lipids) are symmetrically distributed between the inner and the outer leaflet of a liposome membrane (upper part). Upon addition of membrane-impermeable dithionite, NBD-lipids in the outer leaflet are reduced to non-fluorescent 7-amino-2-1,3-benzoxadiazol-4-yl phospholipids (ABD-lipids), resulting in a theoretical loss of 50% in fluorescence. When liposomes are reconstituted in the presence of an active glycerophospholipid scramblase (bottom part), treatment by dithionite results in a theoretical loss of 100% in fluorescence due to NBD-lipid scrambling between the inner and the outer leaflets. (**b**) Coomassie-stained SDS-PAGE gel of purified LacY. Molecular mass markers are indicated in kDa. (**c**–**e**) NBD-lipid scrambling in LacY proteoliposomes. Liposomes containing trace amounts of NBD-PC (**c**), N-NBD-PE (**d**) or NBD-PI (**e**) were reconstituted in the absence or presence of purified LacY using a protein to phospholipid ratio of 0 (grey trace), 0.5 (blue trace) or 2 (red trace) mg/mmol. Dithionite was added (indicated by arrows) and fluorescence was recorded continuously for 8 min. All traces represent mean values from at least three independent experiments. (**f**) Dependence of fluorescence reduction against the protein to phospholipid ratio. The plateaus of fluorescence reduction in NBD-PC, N-NBD-PE and NBD-PI in proteoliposomes reconstituted with different amounts of LacY and plotted against the protein to phospholipid ratio were determined. The data points are from three independent experiments and fit into a one-phase exponential function.

**Figure 2 biology-12-01367-f002:**
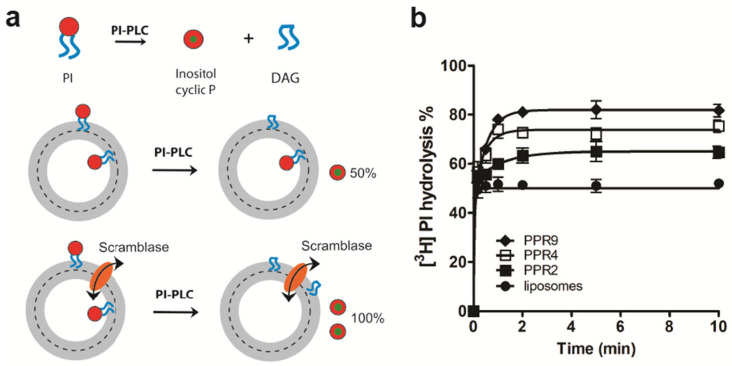
LacY scrambles [^3^H]PI in proteoliposomes. (**a**) Schematic representation of the scramblase assay. [^3^H]PI, with the label being located on the inositol, is distributed symmetrically between the inner and the outer leaflets of a liposome membrane. Upon addition of PI-PLC, [^3^H]PI in the outer leaflet is cleaved into [^3^H]inositol 1,2-cyclic phosphate (inositol cyclic P) and diacylglycerol (DAG), resulting in a theoretical release of 50% of [^3^H]inositol 1,2-cyclic phosphate. In proteoliposomes containing a scramblase, [^3^H]PI from the inner leaflet is translocated to the outer leaflet, where it becomes accessible to cleavage by PI-PLC. The extent of [^3^H]PI hydrolysis in scramblase-containing proteoliposomes is expected to reach 100% theoretically. (**b**) LacY scrambles natural PI. Liposomes containing trace amounts of [^3^H]PI were reconstituted in the absence or presence of purified LacY using increasing protein to phospholipid ratios (PPRs). At the indicated times, the reaction was stopped by adding trichloroacetic acid followed by cytochrome c, and the supernatant containing [^3^H]inositol 1,2-cyclic phosphate/[^3^H]inositol 1-phosphate was collected using centrifugation. The extent of [^3^H]PI hydrolysis was measured against 100% hydrolysis in liposomes disrupted with 0.9% (*w*/*v*) Triton X-100. The data are mean values ± standard deviations from three independent experiments. The data points were fit to a one-phase exponential function for mock-reconstituted liposomes and a two-phase exponential function for proteoliposomes.

**Figure 3 biology-12-01367-f003:**
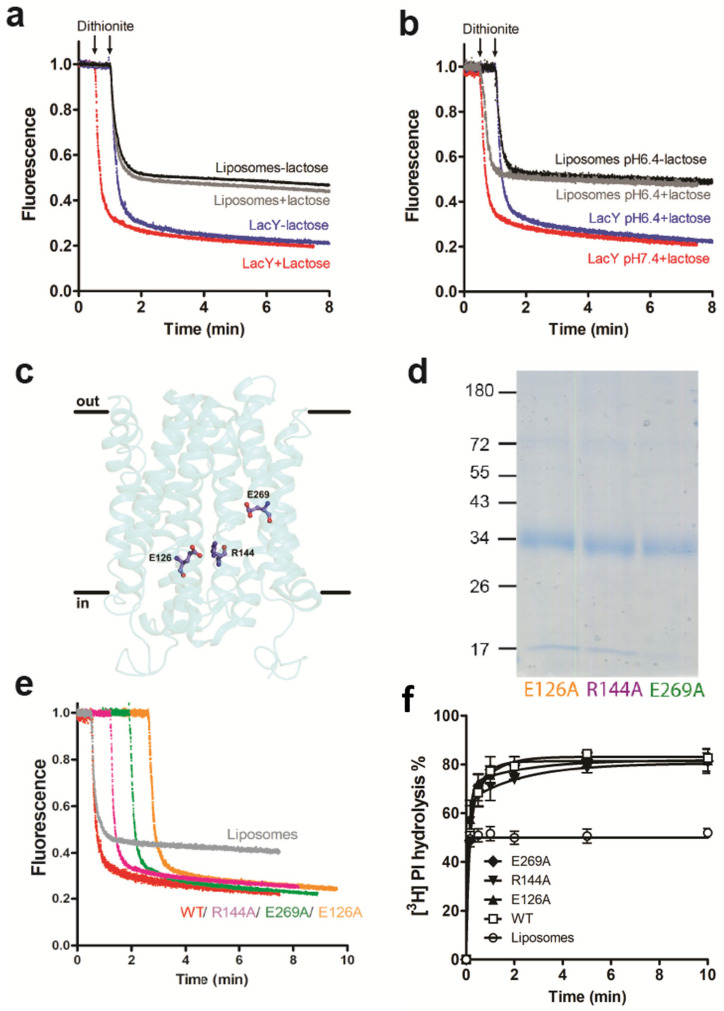
Scramblase activity of LacY is independent of H^+^/lactose transport activity. (**a**) NBD-PC scrambling activity of LacY in the presence and absence of lactose. Liposomes containing trace amounts of NBD-PC were reconstituted with purified wild-type LacY using a PPR of 1 mg/mmol in the absence (blue trace) or presence (red trace) of 1 mM lactose. Dithionite was added after 1 min and fluorescence was recorded continuously for 8 min. For clarity, the trace representing NBD reduction in the presence of lactose (red trace) is shifted to the left by 0.5 min. Traces represent mean values from three independent experiments and show LacY proteoliposomes (blue and red traces) and liposomes mock-reconstituted in the absence (black trace) or presence (grey trace) of lactose. (**b**) NBD-PC scrambling activity of LacY at lower pH. Liposomes containing trace amounts of NBD-PC were reconstituted with purified LacY using a PPR of 1 mg/mmol at pH 7.4 (red trace) or pH 6.4 (blue trace) in the presence of 1 mM lactose. Dithionite was added after 1 min and fluorescence was recorded continuously for 8 min. Traces represent mean values from at least three independent experiments and show LacY proteoliposomes (blue and red traces) and mock-reconstituted liposomes in the absence (black trace) or presence (grey trace) of lactose. For clarity, the traces representing NBD reduction in proteoliposomes at pH 7.4 and mock-treated liposomes in the presence of lactose (grey trace) are shifted to the left by 0.5 min. (**c**) Side view of the LacY structure indicating the locations of the mutated amino acids E126, R144 and E269. (**d**) Coomassie-stained SDS-PAGE gel of purified LacY mutants. Molecular mass markers are indicated in kDa. (**e**) NBD-PC scramblase activity of LacY mutants with dead lactose transport activities. Liposomes containing trace amounts of NBD-PC were reconstituted with wild-type LacY (WT; red trace) or LacY mutants E126A (orange trace), R144A (purple trace) or E269A (green trace), using a PPR of 1 mg/mmol. Dithionite was added after 1 min and fluorescence was recorded continuously for 8 min. For clarity, traces representing NBD reduction in the mutant LacY proteoliposomes are shifted to the right by 0.5 min increments. NBD reduction in mock-reconstituted liposomes is shown by the grey trace. All traces represent mean values from at least three independent experiments. (**f**) LacY mutants E126A, R144A and E269A scramble natural PI. Liposomes containing trace amounts of [^3^H]PI were reconstituted in the absence or presence of purified LacY mutants using a PPR of 9 mg/mmol. PI-PLC-mediated hydrolysis of [^3^H]PI was determined and analyzed as shown in Figure 2b.

**Figure 4 biology-12-01367-f004:**
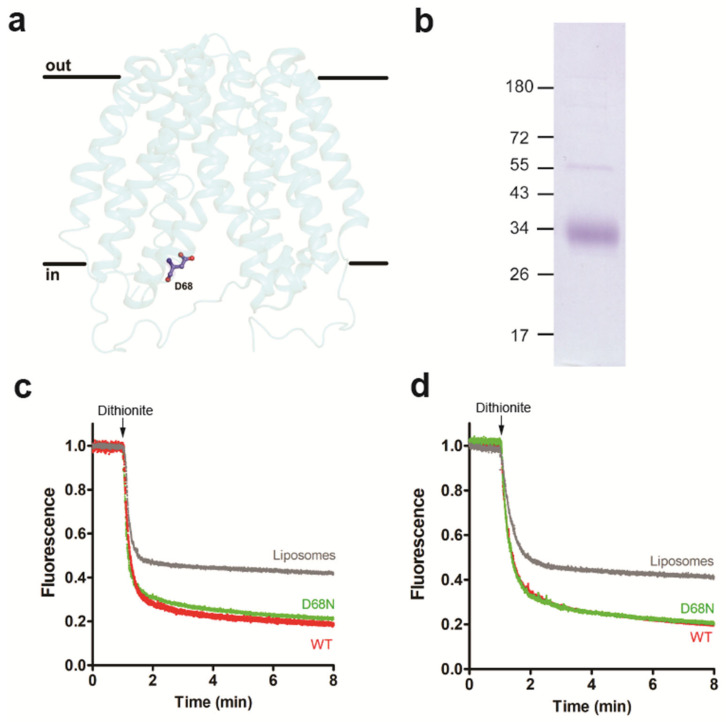
Inward-facing conformation of LacY retains scramblase activity. (**a**) Side view of the structure of LacY mutant D68N. (**b**) Coomassie-stained SDS-PAGE gel of purified LacY mutant D68N. Molecular mass markers are indicated in kDa. (**c**,**d**) Glycerophospholipid scramblase activity of LacY mutant D68N. Liposomes containing trace amounts of NBD-PC (**c**) or N-NBD-PE (**d**) were reconstituted with wild-type LacY (red traces) or LacY mutant D68N (green traces) using a PPR of 1 mg/mmol or mock-reconstituted (grey trace). Dithionite was added after 1 min, and fluorescence was recorded continuously for 8 min. All traces represent mean values from at least three independent experiments.

**Figure 5 biology-12-01367-f005:**
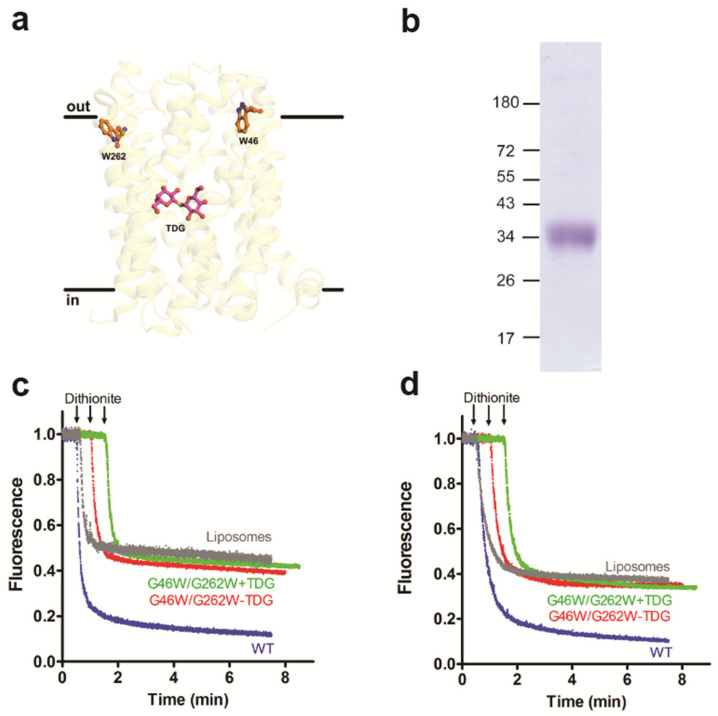
Outward-facing conformation of LacY phospholipid lacks scramblase activity. (**a**) Side view of the structure of LacY mutant G46W/G262W. The binding site of TDG is indicated. (**b**) Coomassie-stained SDS-PAGE gel of purified LacY mutant G46W/G262W. Molecular mass markers are indicated in kDa. (**c**,**d**) Glycerophospholipid scramblase activity of LacY mutant G46W/G262W. Liposomes containing trace amounts of NBD-PC (**c**) or N-NBD-PE (**d**) were reconstituted with wild-type LacY (WT; blue traces) or LacY mutant G46W/G262W in the absence (red traces) or presence (green traces) of 1 mM TDG using a PPR of 2 mg/mmol. Dithionite was added after 1 min, and fluorescence was recorded continuously for 8 min. For clarity, traces representing NBD reduction in LacY mutant G46W/G262W proteoliposomes are shifted to the right by 0.5 min increments. NBD reduction in mock-reconstituted liposomes is shown by the grey trace. All traces represent mean values from at least three independent experiments.

**Figure 6 biology-12-01367-f006:**
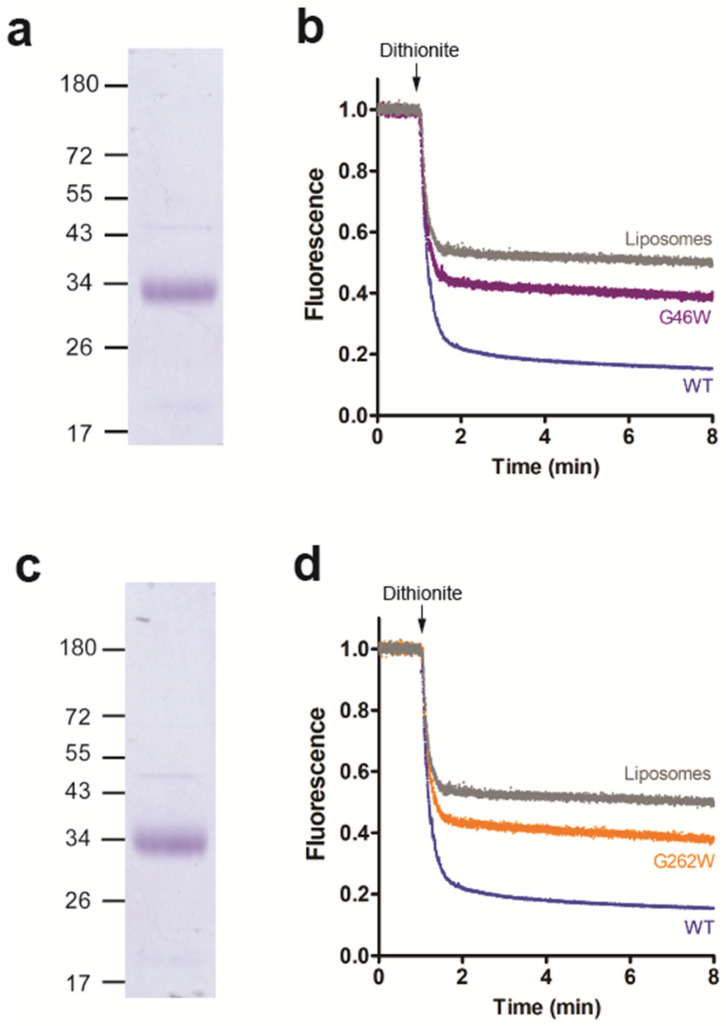
Residues G46 and G262 are important for glycerophospholipid scrambling activity. (**a**,**c**) Coomassie-stained SDS-PAGE gel of purified LacY mutants G46W (**a**) and G262W (**c**). (**b**,**d**) NBD-PC scramblase activity of LacY mutants G46W (**b**) and G262W (**d**). Liposomes containing trace amounts of NBD-PC were reconstituted with wild-type LacY or LacY mutants G46W or G262W using a PPR of 1 mg/mmol or mock-reconstituted. Dithionite was added after 1 min, and fluorescence was recorded continuously for 8 min. All traces represent mean values from at least three independent experiments.

## Data Availability

Not applicable.

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
