# Peer review of "Lactose Permease Scrambles Phospholipids"

_biology, 2023, doi:10.3390/biology12111367_

Round 1
Reviewer 1 Report
In this manuscript, the authors have revealed LacY as a new scramblase. LacY was initially identified as a protein found in the cell membrane of bacteria, particularly in Escherichia coli and related organisms. It plays a crucial role in the transport of lactose into the bacterial cell, which is essential for the utilization of lactose as a carbon source for energy and growth.
Authors have also revealed the connections between its two functions: lactose transport and phospholipids transport. Importantly, these two functions do not rely on each other. Furthermore, this manuscript has identified the specific amino acids responsible for the scrambling function.
The data presented in the manuscript are solid enough to support their conclusions.
Minor comments: 1. It will be great if the authors could describe more about the future applications of this study. 2. Editing of English is required. 3. Please include more comments to discuss the potential shortage of this study and how the authors plan to address this.
Moderate editing of English language required.
Reviewer 2 Report
In this manuscript “Lactose permease scrambles phospholipids” by Lei Wang and Peter Bütikofer, the authors claimed that LacY catalyzes the rapid translocation of labeled glycerophospholipids across proteoliposome membrane bilayer. This conclusion is solely based on the observed symmetric distribution of labeled lipids between the inner and outer leaflets of LacY liposomes reconstituted in the presence of the labeled lipids. Further, this “scrambling activity” is independent of the LacY transport activity or pH changes, but it correlates with LacY conformation. The fluorescent quenching experiments were well carried out but the design and result interpretation had problems.
Scramble activity means the ability to perform the translocation of lipids from inner to outer leaflets. The observed LacY-dependent asymmetric distribution of labeled lipids, either using
Fluorescence (NBD)- or [3H]-labeled glycerophospholipids, is a very interesting novel observation. The fluorescent quenching using the membrane-impermeant dithionite in the presented experiment can report the topological distribution of NBD lipids. This experiments in fig. 2 do not contain the information of lipids translocation process from inner to outer leaflets.
On the other hand, it is conceivable that the quenching rate should be faster than the translocation rate. If the translocation happened during the 8-min trace, one would expect a bi-phase, but all results showed a one-phase curve, which did not support two pools of NBD lipids.
Is that possible that LacY might pick up the soluble NBD lipids at its periplasmic side during the reconstitution, which can alter the distribution of NBD lipids in the first place when the liposomes were formed? Other experiment designs to detect the translocation processing are needed. The addition of the labeled lipids during the trace seems useful for monitoring the scramble activity if any.
LacY favored the inward-facing state.
2w LacY favored an outward-facing state.
D58N mutant conformation is undetermined.
Overall, this study lacks data to support the major conclusion, but the presented data apparently supported the notion that LacY altered the symmetry of lipid distribution.
Reviewer 3 Report
The manuscript by Wang and Bütihofer is a short, single idea manuscript. Authors report that Lactose permease (LacY) from Escherichia coli demonstrates flippase activity. The experiment is straightforward and conclusions follow the results evidently. Nothing is to add to the Results section. At the same time the manuscript is lacking Discussion. I advise authors to extend the manuscript with discussion of possible mechanism of lipid flipping with the help of LacY.
Round 2
Reviewer 2 Report
The author failed to rebut, instead providing a list of literature. Several points I am re-emphasizing here:
- The observed Asymmetric distribution of NBD-PC with LacY-liposomes is clearly documented. This can support publication, only if deleting the claim for the LacY scramblase activity since a lack of supporting data.
- Both assays were well carried out but the authors reported this "end-point" data as scramblase activity (translocation of lipids). There is no data to show the PROCESS of lipid translocation. The end-point data could result from scramblase activity, but it can also stem from another mechanism. In this case, the kinetics of the quenching curve does not support that lipid translocation is involved.
- Your data actually oppsited your claim. If LacY is a scramblase, the quenching curves should reach the same level regardless of LacY concentration. LacY concentration can only change the rates of translocation. But in your figures, it changes both rate and level. If the scramblase activity existed, why stop working?
- Per my knowledge, the quenching reaction should be faster than the lipid translocation. All the data points only showed the quenching reaction, no lipid translocation.
In addition, this claimed activity is independent of LacY activity. Then, what is the driving force for the high energy-consuming process? By the way, E. coli membranes do even not have PC.
